# Gram-Negative Endogenous Endophthalmitis: A Systematic Review

**DOI:** 10.3390/microorganisms11010080

**Published:** 2022-12-28

**Authors:** Giorgio Tiecco, Davide Laurenda, Alice Mulè, Stefania Arsuffi, Samuele Storti, Manuela Migliorati, Alessandro Boldini, Liana Signorini, Francesco Castelli, Eugenia Quiros-Roldan

**Affiliations:** 1Department of Clinical and Experimental Sciences, Unit of Infectious and Tropical Diseases, University of Brescia and ASST Spedali Civili di Brescia, 25123 Brescia, Italy; 2Department of Neurological and Vision Sciences, Eye Unit, University of Brescia and ASST Spedali Civili di Brescia, 25123 Brescia, Italy; 3Unit of Infectious and Tropical Diseases, ASST Spedali Civili di Brescia, 25123 Brescia, Italy

**Keywords:** endophthalmitis, endogenous endophthalmitis, endogenous bacterial endophthalmitis, EBE, Gram-negative, *Klebsiella*, *Pseudomonas*, *Escherichia coli*, systematic review, review

## Abstract

**Background**: Gram-negative bacteria are causative agents of endogenous endophthalmitis (EBE). We aim to systematically review the current literature to assess the aetiologies, risk factors, and early ocular lesions in cases of Gram-negative EBE. **Methods**: All peer-reviewed articles between January 2002 and August 2022 regarding Gram-negative EBE were included. We conducted a literature search on PubMed and Cochrane Controlled Trials. **Results**: A total of 115 studies and 591 patients were included, prevalently Asian (98; 81.7%) and male (302; 62.9%). The most common comorbidity was diabetes (231; 55%). The main aetiologies were *Klebsiella pneumoniae* (510; 66.1%), *Pseudomonas aeruginosa* (111; 14.4%), and *Escherichia coli* (60; 7.8%). Liver abscesses (266; 54.5%) were the predominant source of infection. The most frequent ocular lesions were vitreal opacity (134; 49.6%) and hypopyon (95; 35.2%). Ceftriaxone (76; 30.9%), fluoroquinolones (14; 14.4%), and ceftazidime (213; 78.0%) were the most widely used as systemic, topical, and intravitreal anti-Gram-negative agents, respectively. The most reported surgical approaches were vitrectomy (130; 24.1%) and evisceration/exenteration (60; 11.1%). Frequently, visual acuity at discharge was no light perception (301; 55.2%). **Conclusions**: Gram-negative EBEs are associated with poor outcomes. Our systematic review is mainly based on case reports and case series with significant heterogeneity. The main strength is the large sample spanning over 20 years. Our findings underscore the importance of considering ocular involvement in Gram-negative infections.

## 1. Introduction

Endogenous endophthalmitis (EE) is a rare but devastating complication of bloodstream infections found in less than 0.5% of patients with fungemia and 0.04% of patients with bacteraemia [1]. When left untreated, endogenous bacterial endophthalmitis (EBE) can damage the eye’s structures, leading to visual impairment and even blindness [2]. From any possible source of infection, the aetiological agent may spread through the bloodstream across the blood–retinal barrier (BRB), eventually reaching the eye structures [2]. Nearly 10% of EBE worldwide are caused by *Staphylococcus aureus* (*S. aureus*), which is also the leading cause of EBE in the USA and Europe, comprising approximately 25% of these cases [3]. 

*S. aureus* itself can alter BRB tight junctions by disrupting the expression and/or organisation of the ZO-1 protein [4]. In other words, *S. aureus* possesses the ability to cause EBE regardless of pre-existing vascular leakage [5]. However, other bacteria are involved in the aetiology of EBE: *Streptococcus* spp. (including *viridans group*, *S. pneumoniae*, *Streptococcus milleri group* and group A and B streptococci), and Gram-negative pathogens such as *Escherichia coli* and especially *Klebsiella pneumoniae* represent important causes of EE [1]. A higher rate of endophthalmitis has been reported in patients with hypervirulent *K. pneumoniae* (hvKp) bacteraemia associated with liver abscesses and prostate involvement [6]. Unlike *S. aureus*, no intrinsic pathogenic activity has been demonstrated for these bacteria. However, predisposing conditions that might cause damage to the BRB do exist. Diabetes mellitus ranks highest among the comorbidities related to EBE. It is associated with 33% of cases and causes significant permeability alterations in the BRB [4]. Studies in animal models have suggested that an increased BRB permeability could contribute to an increment in bacterial transmigration from the bloodstream into the eye [5].

It is uncertain whether it is necessary to conduct ocular screening for Gram-negative EBEs in day-by-day clinical practice. Literature on EBE consists mostly of case series or single case reports and only a few analyses on humans investigate risk factors for BRB alteration in cases of ocular involvement secondary to Gram-negative bloodstream infections [2]. Lastly, several primary focal ocular lesions might break into and seed the vitreous causing EBE, but these pathogenetic aspects are not well defined, especially for Gram-negative bacteria [7,8,9]. Our primary aim is to systematically review the current literature to properly assess the risk factors, main aetiologies, and early ocular lesions in case of EBE due to Gram-negative bacteria.

## 2. Methods

Our methods meet the Preferred Reporting Items for Systematic Reviews and Meta-Analysis (PRISMA) updated guideline for systematic review stated in 2020 [10].

### 2.1. Eligibility Criteria

All articles published in peer-reviewed medical journals between January 2002 and August 2022 regarding EBE during Gram-negative infection were included. We excluded articles regarding non-bacterial EE or EE secondary to Gram-positive infection papers in which data regarding EBE due to a Gram-negative infection were available but impossible to extrapolate. Articles published in non-English languages, pre-print or ahead of print analysis, pre-clinical studies, reviews, systematic reviews, and metanalysis were excluded too.

### 2.2. Information Sources and Search Strategy

With the assistance of a professional medical librarian at our institution, we determined our process for the literature search. We consulted the United States National Library of Medicine, PubMed (last accessed August 2022), and the Cochrane Controlled Trials (last accessed August 2022). References for this review were identified with the following research terms combination: “endogenous endophthalmitis” AND “gram negative”. As the term “gram negative” was often taken for granted in the title or abstract of articles regarding EBE caused by *Klebsiella* spp., *E. coli*, or *Pseudomonas* spp., we decided to expand our search strategy by also including these combinations: “endogenous endophthalmitis” AND “Klebsiella” OR “Pseudomonas” OR “Escherichia coli” which are considered as the Gram-negative bacteria predominantly involved in EBE [11,12]. 

### 2.3. Selection and Data Collection Process

A team of 7 resident doctors in Infectious and Tropical Diseases of the University of Brescia, Italy, read the abstract of each scientific work and independently selected the articles according to the established criteria (SA, DL, MM, AM, SS, GT). A Professor in Infectious and Tropical Diseases and an Ophthalmologist of the ASST Spedali Civili di Brescia, Italy, revised the included and the rejected papers. Then, resident doctors formed two teams: the first one (SA, AM, SS, GT) collected data by considering each selected article full text, while the second group (DL, MM) created a thorough database to revise, compare and synthesise data. An ophthalmologist revised the collection and synthesis of the ophthalmologic data. No automated tools were used.

### 2.4. Data Items

For each selected article, we collected information regarding the number of patients with an EBE due to a Gram-negative infection, their demographic data (age, sex, and ethnicity), comorbidities, and the number of eyes involved, specifying (when available) if a right/left eye was affected. Aetiological data (Gram-negative bacteria involved, culture type and source of infection), as well as initial ocular lesions and visual acuity were reported. Furthermore, we assessed ocular and systemic complications, medical therapy (anti-Gram-negative topical, intravitreal, or systemic antibiotics along with the addition of steroids) and the eventual surgical approach employed. We reported the general and ocular outcome and of a follow-up visit within the 12 months after discharge had been performed. Missing or unclear data were reported as “non-available”. Similarly, we considered “non-available” data regarding EBE due to a Gram-negative infection but impossible to extrapolate because included in more comprehensive studies concerning endophthalmitis in general. 

### 2.5. Synthesis Methods

All the collected data were reported in a single table that was revised by an independent group. Every column was specifically associated with a different item. In the case of columns with less than 5 records, a grouping of the result was performed: i.e., in the case of poorly represented bacterial species, we preferred grouping them under an “other gram-negative aetiology” column. We limited our study to a descriptive analysis of our findings due to the wide heterogenicity of the articles selected. The percentage calculation was performed in consideration of the number of data available for each specific item. No models to identify the presence and extent of statistical heterogeneity or sensitivity analyses to assess the robustness of the synthesised results were performed.

### 2.6. Bias and Certainty Assessment

This is a systematic review for which a descriptive analysis has been performed due to the wide heterogenicity of the selected articles. Risk of bias or certainty (or confidence) in the body of evidence was not assessed. 

## 3. Results 

### 3.1. Study Selection and Search Results

A total of 154 papers were identified through our search. We excluded 19 duplicate articles. A further eight analyses were removed as five were systematic reviews or meta-analyses [3,13,14,15,16] and three were pre-clinical sciences papers [17,18,19]. Moreover, two studies were excluded because the full text was unavailable [20,21] and one paper was removed due to a lack of data regarding the peer-revision process of the journal [22]. The remaining 124 articles were assessed for eligibility. Seven were excluded as data regarding EBE due to a Gram-negative infection were available, but impossible to select [23,24,25,26,27,28,29] and two further analyses were removed as they did not fully meet the inclusion criteria [30,31]. Eventually, 115 studies were included, as shown in the following flow diagram (Figure 1). 

Most studies were case reports (57, 49.6%) and retrospectively non-randomized (46, 40%). Regarding the geographic distribution of the studies, 58.1% of the articles were from Asia, 13.2% were from Europe, and 9.6% were from America. Study characteristics, patient comorbidities, and aetiologic data included are summarized in Table 1 [32,33,34,35,36,37,38,39,40,41,42,43,44,45,46,47,48,49,50,51,52,53,54,55,56,57,58,59,60,61,62,63,64,65,66,67,68,69,70,71,72,73,74,75,76,77,78,79,80,81,82,83,84,85,86,87,88,89,90,91,92,93,94,95,96,97,98,99,100,101,102,103,104,105,106,107,108,109,110,111,112,113,114,115,116,117,118,119,120,121,122,123,124,125,126,127,128,129,130,131,132,133,134,135,136,137,138,139,140,141,142,143,144,145,146].

### 3.2. Results of Synthesis

A total of 591 patients were included. Considering the available demographic data, patients were prevalently Asian (98/120, 81.7%), male (302/480, 62.9%), and with a median age of 55.6 years old. As shown in Table 2, the most common comorbidities identified were diabetes mellitus (231, 55%), hypertension and other cardiovascular diseases (79, 18.8%), renal diseases (21, 5.0%), and malignancies (19, 4.5%). 

Overall, 592 infected eyes were involved, with a higher percentage of monocular EBE (429, 83.1%) than binoculars (67, 13%). Focusing on the available data, right eyes (277, 53.7%) were more involved than left eyes (239, 46.3%). 

A total of 772 Gram-negative bacteria were included. As shown in Figure 2, *Klebsiella pneumoniae* (510, 66.1%) was the most common pathogen isolated in the case of Gram-negative EBE, followed by *Pseudomonas aeruginosa* (111, 14.4%), *Escherichia coli* (60, 7.8%), and *Haemophilus influenzae* (11, 1.4%). 

These pathogens were isolated both from non-ocular (387, 42.5%) and ocular samples (286, 57.5%). More specifically, the microbiological diagnosis was prevalently performed on vitreous culture (239, 83.6%) and blood cultures (273, 70.5%) when considering the overall number of ocular and non-ocular samples, respectively. As shown in Table 3, liver abscesses (266, 54.5%) represented the predominant primary source of infection of the described EBEs, followed by bloodstream infections/sepsis (116, 23.8%), pneumonia (37, 7.6%), and abdominal infections (37, 7.6%).

Overall, 270 initial ocular lesions were described. Vitreous opacity (134, 49.6%) and hypopyon (95, 35.2%) were the most commonly reported distinctive signs of EBE (Table 4). At the patient’s hospital admission, HM or hand motion (84, 25.4%) and LP or light perception (66, 19.9%) were the most frequently described visual acuities. Ocular complications were uncommon: bulbar atrophy (24, 10.9%), retinal detachment (13, 5.9%), and perforation (12, 5.5%) were the most prevalent. The most frequent systemic complications were septic emboli (4, 2.0%) and central nervous system infections (3, 1.5%).

Few studies reported accurate therapeutic information regarding the antibiotics used and their route of administration. Ceftriaxone (76, 30.9%) was the most widely used anti-Gram-negative systemic antibiotic agent, while fluoroquinolones (14, 14.4%) and ceftazidime (213, 78.0%) were prevalently administered as topical or intravitreal agents, respectively. Only 25 studies appropriately reported antimicrobial therapy’s duration, with an overall mean duration of 39.2 days (range 13 to 84 days). Few patients (69, 28.3%) needed concomitant steroids in addition to the ongoing antimicrobial regimen. Regarding the surgical approach, the most frequently reported techniques were vitrectomy (130, 24.1%) and evisceration/exenteration (60, 11.1%).

Regarding the clinical outcomes at the end of the hospitalization, most patients were discharged (238, 85%), and mortality was recorded in only 15 cases. A higher percentage of NLP or no light perception (301, 55.2%) was reported as final ocular outcome compared to the initial visual acuity assessed. Only 25 studies reported follow-up information, and only five relapses occurred within 12 months after discharge.

## 4. Discussion

This systematic review estimates the clinical and epidemiological impact of Gram-negative EE, by analysing over a hundred papers spanning 20 years. EBEs, defined as the infection of intraocular tissues resulting from the hematogenous spread of bacteria to the eye, are both a diagnostic and therapeutic challenge for ophthalmologists and infectious diseases specialists [67]. Gram-negative EBEs are an undoubtedly consistent clinical reality associated with poorer outcomes due to the production of endotoxins and the phagocytosis-resistant capsules conferring greater virulence [67,147].

Our search shows that, in East Asian nations, many EBEs are caused by Gram-negative bacilli, including *Klebsiella pneumoniae* and *Escherichia coli* [1]. Studies from Singapore and Taiwan showed that up to 70% of the organisms isolated from patients with EBE were Gram-negative [67]. Similarly, other analyses reported frequencies ranging from 22.2% to 77.1%, considering Gram-negative as causative agents of EBE [147]. Interestingly, *Klebsiella* was found to be the most common causative organism (31.7%-87.6%) followed by *Pseudomonas aeruginosa* [147].

In our systematic review, *Klebsiella pneumoniae* was the most common pathogen isolated, while liver abscesses represented the primary source of infection. Indeed, the association of liver abscesses with *Klebsiella* as the causative organism is observed worldwide, especially when considering hypervirulent strains (hvKp) [6]. Although the mortality rate of hvKp liver abscess is relatively low compared to that associated with pyogenic liver abscesses caused by bacteria other than *K. pneumoniae*, hvKp infection can lead to metastatic complications that cause significant morbidity such as, for instance, EBE [15]. Most hvKp infections are community-acquired, often afflicting individuals without any predisposing medical condition [15]. The incidence of hvKp infections seems to be rising both in Asia and Europe, and this can be explained by the rates of hvKp-carriers that range from 19% to the alarming percentage of 88% of healthy Chinese adults [15,148]. In a recent systematic review, 1 out of 22 patients with *Klebsiella pneumoniae* pyogenic liver abscess was found to develop EBE. This is explained by the K antigen, a capsular polysaccharide and a well-established virulence factor that makes K1 serotype infection an independent risk factor for the development of EE [15]. 

Although cases of EBE have been reported in otherwise healthy and immunocompetent people, EBEs are frequently associated with many systemic risk factors, including chronic immune-compromising illnesses, immunosuppressive diseases or therapies, recent invasive surgery or gastrointestinal procedures, hepatobiliary tract infections, and intravenous drug use [67]. Diabetes is the primary underlying condition associated with EBE (46–63.86%) in Asia. Considering the current scenario of the COVID-19 pandemic, the heavy use of systemic corticosteroids can predispose patients to the subsequent development of EBE via steroid-induced diabetes [147,149]. Although the pathogenic mechanism is poorly understood, it is known that poor glycaemic control might impair neutrophilic hepatic Kupffer cells’ phagocytosis against the bacterial infiltrators arriving with portal blood [15]. A recent review of case series published between 2011 and 2020 stated that while diabetes mellitus remains one of the medical conditions most frequently associated with EBE, malignancies and intravenous drug use represent significant risk factors too [11]. Malignancies were thought to be prevalently associated with endogenous mould endophthalmitis, where *Aspergillus* spp. and *Fusarium* spp. were the major pathogens involved [1]. However, malignancies have also been found to be a risk factor in the case of *Streptococcus* spp., *Pseudomonas* spp., and *Candida* spp. endogenous endophthalmitis [11]. In our systematic review, while malignancies are well represented, a small percentage of intravenous drug use is reported. This is consistent with the current literature, since the majority of EBE in people who inject drugs are caused by Gram-positive rather than Gram-negative agents [150].

In our study, vitreous opacity and hypopyon were EBE’s most described initial lesions. However, eye redness (91, 33.8%) alone or together with other ocular signs was commonly reported. This finding, together with the not always severely compromised visual acuity, enlightens the need for ophthalmologists to maintain high suspicion for EBE in patients with intraocular inflammation and significant medical comorbidities [67]. Patients with EBE usually present acutely, complaining only about decreased vision and eye pain [1]. Systemic complications or more alarming local signs such as hypopyon or vitritis might be absent during the initial evaluation [1]. 

The treatment of EBE should include both ocular and systemic therapy. This is a pharmacokinetic consequence, since most antimicrobial agents have a poor penetration capacity into the avascular vitreous cavity when parenterally administered [67]. Therefore, intravitreal injections are the treatment of choice for EBE. In line with our findings, the most commonly used antimicrobials for empiric treatment are third generation cephalosporines for Gram-negative microorganisms, followed by amikacin and gentamicin, which were mostly used in combination regimens [151]. The notorious Endophthalmitis Vitrectomy Study, a randomized clinical trial conducted between 1991 and 1994, stated that 89.5% of Gram-negative organisms causing endophthalmitis were susceptible to both amikacin and ceftazidime [152]. Although the emergence of multidrug-resistant bacteria is a global issue, the antibiotic susceptibility patterns of Gram-negative bacteria from vitreous isolates have not significantly changed in the United States [152].

The role of additional steroids in EBE management is controversial. A recent study in the Cochrane Library states that the currently available evidence on the effectiveness of adjunctive steroid therapy versus antibiotics alone in managing acute endophthalmitis after intraocular surgery is inadequate [153]. A combined analysis of a very limited number of studies suggests that adjunctive steroids might provide a higher chance of having a better visual outcome at three months [153]. Moreover, another study shows a higher rate of enucleation/evisceration in patients who did not receive steroid therapy [147]. These controversial findings match the differences in clinical approaches to EBE management and treatment revealed by our systematic review, where just a few patients were treated with steroids in addition to the ongoing antimicrobial regimen.

Adequate source control is often warranted in the case of EBE. Surgical intervention is generally recommended for patients infected with virulent organisms, with bilateral involvement, severe vitreous involvement, and progressive worsening [67]. Our systematic review shows that vitrectomy is the most often used surgical procedure as it helps in removing infectious organisms, toxins, and inflammatory cells from the vitreous cavity, thus leading to a better diffusion of antibiotics and a faster recovery [147]. Vitrectomy has several clinical and diagnostic implications: it might save eyes with EBE and restore vision while also providing a higher diagnostic yield compared to a vitreous biopsy, thus helping identify the causative organism [147]. 

Prognosis is poor in cases of Gram-negative EBE. Despite aggressive therapy, often necessitating surgical intervention, the poor clinical outcome in the case of EBE might be related to a delay in the diagnosis and treatment or the absence of worldwide shared guidelines [15]. Although several factors are associated with the visual outcome, a central role seems to be played by the pathogen involved. A recent study reported that very poor visual acuity (20/400 or worse) is associated with several Gram-negative pathogens such as *H. influenzae* (69%), *Serratia* spp. (70%), and *Pseudomonas* spp. (92%) [1]. Although not uniformly observed across all studies, it has been hypothesized that Gram-negative EE’s poorer outcome could be linked to both Gram-negative endotoxin production and the presence of a phagocytosis-resistant capsule [147].

The findings of this systematic review should be seen in the light of some limitations. First, our research strategy includes a selection bias that cannot be eliminated. Indeed, by including the three main EBE aetiologies, “Klebsiella”, “Escherichia coli”, and “Pseudomonas” in the initial search, it is subordinate that their prevalence will be found to be higher. However, the selection of these species allowed the authors to consider more papers that would have otherwise been wrongfully excluded. Secondly, our systematic review is based on many case reports and case series, with only one prospective study and no RCTs. Consequently, the inclusion of retrospective studies describing aggregate data makes it hard to select data for each patient individually. Lastly, the heterogeneity of the studies included in the absence of methods to assess the risk of bias or certainty in the body of evidence restricted our review to descriptive analysis. Therefore, we limited our comprehensive analysis to a descriptive evaluation of the past 20 years’ literature on Gram-negative EBE. On the other hand, the main strength of this systematic review is the large sample size. In addition, as it was noted during the search phase that most of the scientific output on the subject is produced by ophthalmologists, this systematic review presented Gram-negative EBEs from an infectious diseases specialist’s point of view.

## 5. Conclusions

Although the literature on EBE mainly comprises case series or single case reports, Gram-negative EBE is an undoubtedly consistent clinical reality associated with poorer outcomes due to virulence and pathogenetic aspects of the Gram-negative’s structure. *Klebsiella pneumoniae* is the most common causative pathogen in Gram-negative EBE, especially in the Asian population or diabetic people. Although in our study vitreous opacity and hypopyon were the most often described initial lesions of EBE, eye redness alone or together with other ocular signs was commonly reported. This enlightens the need for ophthalmologists to maintain high suspicion for EBE in patients with intraocular inflammation and significant medical comorbidities. Our findings underscore the importance of considering ocular involvement in the case of Gram-negative infections. In light of an ageing population and considering the concerning phenomenon of Gram-negative antimicrobial resistance, EBEs’ appropriate management remains an open challenge for both ophthalmology and infectious disease specialists. 

## Figures and Tables

**Figure 1 microorganisms-11-00080-f001:**
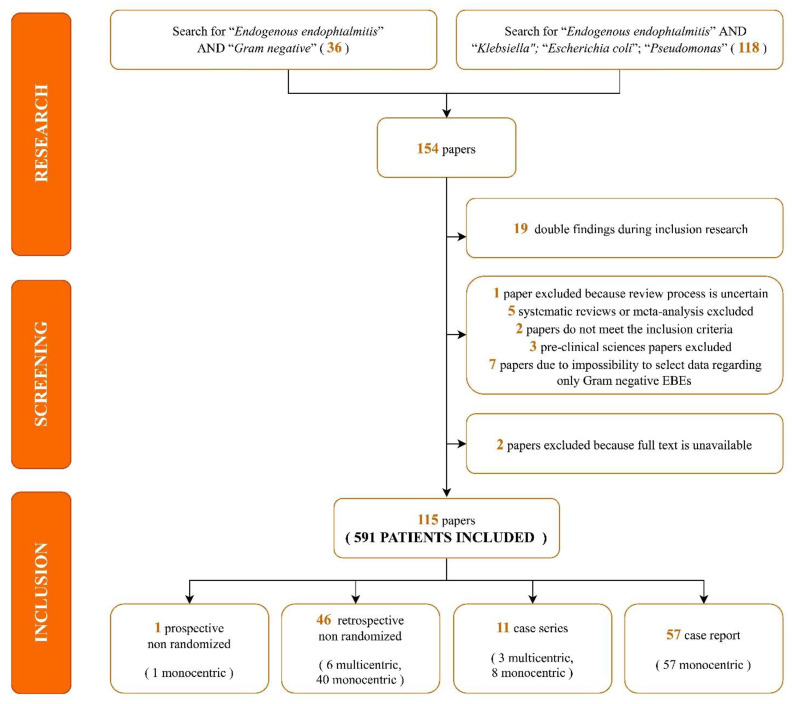
Search strategy and selection process flow-chart.

**Figure 2 microorganisms-11-00080-f002:**
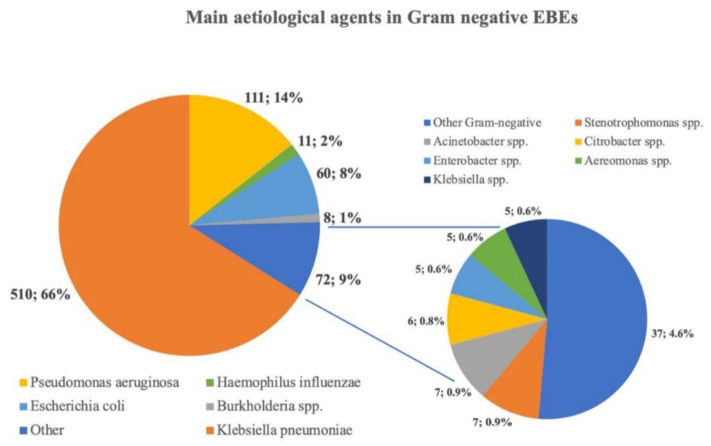
Prevalent aetiology found in this systematic review. Each pathogen included in the “Other” group represents less than 1% of the aetiology reported. *Klebsiella* spp. = *Klebsiella* other than *K. pneumoniae*.

**Table 1 microorganisms-11-00080-t001:** Summary table regarding study characteristics, aetiologic data and patients’ comorbidities.

Country	Patients	Aetiology	Source of Infection	Comorbidities	N° of Eyes (n° RE, n° LE)	Monocular/Binocular	Surgery	Reference
Taiwan	48	*Klebsiella pneumoniae* 31 *Pseudomonas aeruginosa* 11 *Escherichia coli* 2	Pneumonia 2 Other 20	Diabetes mellitus 32 Hypertension 21 Renal disease 7 Cirrhosis 6 HIV 2 Malignancy 1	48 (RE 24 LE 24)	Monocular 48	Vitrectomy 6 Enucleation 18	[32]
Palestine	2	*Klebsiella pneumoniae* 2	NA	Diabetes mellitus 2 Malignancy 1	2 (RE 1 LE 1)	Monocular 2	NA	[33]
China	15	*Klebsiella pneumoniae* 13 *Escherichia coli* 1	Liver abscess 8 Other 6 Pneumonia 4	Diabetes mellitus 14	16 (RE 9 LE 7)	Monocular 14 Binocular 1	Evisceration 8	[34]
China	4	*Klebsiella pneumoniae* 4	NA	NA	4 (RE 2 LE 2)	Monocular 4	NA	[35]
Thailand	13	*Klebsiella pneumoniae* 7 Other 4 *Escherichia coli* 1 *Pseudomonas aeruginosa* 1	NA	NA	NA	NA	NA	[36]
India	3	*Klebsiella pneumoniae* 1 *Pseudomonas aeruginosa* 1 *Haemophilus influenzae* 1	NA	NA	NA	NA	NA	[37]
Iran	NA	*Klebsiella pneumoniae* 14 *Escherichia coli* 9 *Pseudomonas aeruginosa* 6	NA	NA	NA	NA	NA	[38]
Australia	NA	*Klebsiella pneumoniae* 9 *Escherichia coli* 1 *Pseudomonas aeruginosa* 1	NA	NA	NA	NA	NA	[39]
India	NA	Other 28 *Pseudomonas aeruginosa* 15 *Escherichia coli* 8 *Haemophilus influenzae* 7 *Klebsiella pneumoniae* 6	NA	NA	NA	NA	NA	[40]
China	10	*Klebsiella pneumoniae* 10	Liver abscess 4 Pneumonia 2 BSI/Sepsis 5 Other 1	Diabetes mellitus 5 Malignancy 2 Cardiovascular disease 1 Other 2 Chronic hepatitis 2	10 (RE 7 LE 3)	Monocular 10	Vitrectomy 3 Evisceration 2 Exenteration 1	[41]
Nepal	2	*Pseudomonas aeruginosa* 2	NA	NA	NA	NA	NA	[42]
China	9	*Klebsiella pneumoniae* 8 Other 1	Liver abscess 16	NA	NA	NA	NA	[43]
Taiwan	NA	*Klebsiella pneumoniae* 44 *Escherichia coli* 9 Other 4 *Pseudomonas aeruginosa* 6 *Haemophilus influenzae* 1	NA	NA	NA	NA	NA	[44]
Thailand	14	*Klebsiella pneumoniae* 9 *Escherichia coli* 3 *Pseudomonas aeruginosa* 2	Pneumonia 1 Other 7	Diabetes mellitus 7 Cirrhosis 2 Hypertension 1 Renal disease 1	14 (RE 9 LE 5)	Monocular 14	Vitrectomy 7 Enucleation 1	[45]
Turkey	2	*Pseudomonas aeruginosa* 1 Other 1	NA	NA	NA	NA	NA	[46]
Hong Kong	19	*Klebsiella pneumoniae* 12	Liver abscess 18 BSI/Sepsis 18 Pneumonia 3 Other 4	Diabetes mellitus 10 Chronic hepatitis 3	24 (RE 10 LE 14)	Monocular 14 Binocular 5	Vitrectomy 3 Evisceration 9	[47]
Taiwan	48	*Klebsiella pneumoniae* 48	Liver abscess 33 Pneumonia 2 Other 6	Diabetes mellitus 34 Hypertension 17 Malignancy 7 Chronic hepatitis 4	10 (RE 2 LE 8)	Monocular 8 Binocular 1	Vitrectomy 2 Enucleation 1 Evisceration 1	[48]
Taiwan	9	*Escherichia coli* 3 Other 3 *Pseudomonas aeruginosa* 2	NA	Diabetes mellitus 4 Hypertension 4 Malignancy 2 Drug abuse 1	58 (RE 27 LE 31)	Monocular 38 Binocular 10	Vitrectomy 18 Enucleation 1 Evisceration 11	[48]
Taiwan	7	*Klebsiella pneumoniae* 3 *Pseudomonas aeruginosa* 3 Other 1	Pneumonia 1 Other 1	Diabetes mellitus 5 Cirrhosis 1	8 (RE 5 LE 3)	Monocular 6 Binocular 1	Vitrectomy 1 Enucleation 1	[49]
Taiwan	14	*Klebsiella pneumoniae* 12 *Escherichia coli* 1 *Pseudomonas aeruginosa* 1	Liver abscess 8 Pneumonia 1 Other 2	Drug abuse 1	14 (RE 10 LE 4)	Monocular 14	NA	[50]
India	8	*Pseudomonas aeruginosa* 6 Other 2	NA	NA	8	Monocular 8	NA	[51]
USA, South Korea	NA	*Klebsiella pneumoniae* 24 *Escherichia coli* 1 *Haemophilus influenzae* 1 Other 1	NA	NA	29	NA	Evisceration 2	[52]
Israel	1	*Escherichia coli* 1	Other 1	NA	NA	NA	NA	[53]
Iran	1	*Pseudomonas aeruginosa* 1	NA	Diabetes mellitus 1	NA	NA	Evisceration 1	[54]
South Korea	8	*Klebsiella pneumoniae* 8	Liver abscess 8	NA	NA	NA	NA	[55]
India	9	*Klebsiella pneumoniae* 1 *Pseudomonas aeruginosa* 3 Other 4	BSI/Sepsis 1	NA	NA	NA	Vitrectomy 3 Evisceration 1	[56]
India	11	*Pseudomonas aeruginosa* 5 Other 3 *Escherichia coli* 2	NA	Diabetes mellitus 4 Gastrointestinal disorders 1 Cirrhosis 1 Chronic hepatitis 1	9 (RE 7 LE 2)	Monocular 9	No surgery	[56]
USA	2	*Pseudomonas aeruginosa* 2	BSI/Sepsis 1	Other 1 Immunosuppression 1 Malignancy 1	2	Monocular 2	Enucleation 2	[57]
South Korea	23	*Klebsiella pneumoniae* 20 *Escherichia coli* 1 *Pseudomonas aeruginosa* 1	Liver abscess 17 Pneumonia 2 Other 1	NA	23 (RE 14 LE 9)	Monocular 23	Vitrectomy 9 Enucleation 1	[58]
India	3	*Pseudomonas aeruginosa* 1 *Haemophilus influenzae* 1 Other 2	NA	NA	NA	NA	NA	[59]
Korea	8	*Klebsiella pneumoniae* 8 *Escherichia coli* 1	Liver abscess 5	Diabetes mellitus 4 Hypertension 3 Renal disease 3 None 2 Cardiovascular disease 1 Gastrointestinal disorders 1 Lung diseases 1 Other 1	8 (RE 5 LE 3)	Monocular 8	Vitrectomy 2 Enucleation 1	[60]
India	14	*Pseudomonas aeruginosa* 8 Other 4 *Klebsiella pneumoniae* 2	BSI/Sepsis 11	Pregnancy complications 3 Gastrointestinal disorders 1	17 (RE 11 LE 6)	Monocular 11 Binocular 3	Vitrectomy 5	[61]
South Korea	8	*Klebsiella pneumoniae* 6 *Escherichia coli* 1 Other 1	NA	NA	8	Monocular 8	NA	[62]
South Korea	30	*Klebsiella pneumoniae* 30	Liver abscess 18 Pneumonia 5 Other 6	Renal disease 2 Diabetes mellitus 12 Cirrhosis 6 Malignancy 3 Alcohol abuse 1	NA	NA	NA	[63]
Taiwan	42	*Klebsiella pneumoniae* 42	Liver abscess 42	Diabetes mellitus 33 Hypertension 18 Gastrointestinal disorders 8 Other 3	53 (RE 24 LE 29)	Monocular 31 Binocular 11	Vitrectomy 9	[64]
Taiwan	8	*Pseudomonas aeruginosa* 8	NA	NA	9 (RE 5 LE 4)	Monocular 7Binocular 1	Evisceration 7	[65]
South Korea	15	*Klebsiella pneumoniae* 7 *Pseudomonas aeruginosa* 7 *Escherichia coli* 1	Liver abscess 3 Pneumonia 2 Other 3	Diabetes mellitus 6 Lung diseases 2 Hypertension 1 Renal disease 1 Chronic Hepatitis 3 Malignancy 1 None 1	15	Monocular 15	Vitrectomy 8 Evisceration 2	[66]
China	1	*Klebsiella pneumoniae* 1	Liver abscess 1	Diabetes mellitus 1	1 (RE 1)	Monocular 1	No surgery	[67]
Taiwan	9	*Klebsiella pneumoniae* 8	BSI/Sepsis 8 Pneumonia 2	Diabetes mellitus 5 Chronic hepatitis 1 Other 3	10 (RE 5 LE 5)	Monocular 8 Binocular 1	Vitrectomy 1	[68]
Singapore	5	*Klebsiella pneumoniae* 5	Liver abscess 5 BSI/Sepsis 5	Diabetes mellitus 4 Hypertension 1	7 (RE 4 LE 3)	Monocular 3 Binocular 2	Vitrectomy 1	[69]
New Zealand	NA	Other 3	NA	NA	NA	NA	NA	[70]
Germany	1	Other 1	NA	NA	1	Monocular 1	NA	[71]
India	2	*Klebsiella pneumoniae* 1 *Pseudomonas aeruginosa* 1	BSI/Sepsis 1	Diabetes mellitus 1 Renal disease 1 Chronic hepatitis 1	2 (RE 1 LE 1)	Monocular 2	Evisceration 1	[72]
Australia	1	*Pseudomonas aeruginosa* 1	Other 1	Diabetes mellitus 1	1 (LE 1)	Monocular 1	Evisceration 1	[73]
Taiwan	45	*Klebsiella pneumoniae* 45	Liver abscess 39 Pneumonia 5 Other 4	NA	NA	NA	NA	[74]
Taiwan	6	*Escherichia coli* 2 Other 2 *Pseudomonas aeruginosa* 1	NA	NA	56 (RE 33 LE 23)	Monocular 34 Binocular 11	Vitrectomy 6	[74]
USA	1	*Klebsiella pneumoniae* 1	BSI/Sepsis 1	Diabetes mellitus 1 Hemopathy 1	1 (RE 1)	Monocular 10	Evisceration 1	[75]
Singapore	10	*Klebsiella pneumoniae* 10	Liver abscess 10	Diabetes mellitus 7 Cardiovascular disease 1	10 (RE 7 LE 3)	Monocular 10	Vitrectomy 3 Evisceration 2	[76]
United Kingdom	7	*Klebsiella pneumoniae* 1 *Escherichia coli* 4 Other 2	Liver abscess 1 Other 5	Diabetes mellitus 4 Other 1 Renal disease 2 Alcohol abuse 1	7 (RE 4 LE 3)	Monocular 7	NA	[77]
South Korea	7	*Klebsiella pneumoniae* 7	Liver abscess 4 BSI/Sepsis 2	Diabetes mellitus 5 Cirrhosis 1	10	NA	Vitrectomy 7	[78]
China	4	*Klebsiella pneumoniae* 4	Liver abscess 4	Diabetes mellitus 2	4 (RE 2 LE 2)	Monocular 4	Evisceration 4	[79]
Spain	1	Other 1	NA	Diabetes mellitus 1	1 (RE 1)	Monocular 1	No surgery	[80]
Spain	1	*Pseudomonas aeruginosa* 1	BSI/Sepsis 1	Immunosuppression 1 Lung diseases 1	1 (RE 1)	Monocular 1	No surgery	[81]
Canada	2	*Klebsiella pneumoniae* 2	BSI/Sepsis 2 Liver abscess 1 Other 1	Diabetes mellitus 1 Gastrointestinal disorders 1	4 (RE 2 LE 2)	Binocular 2	Enucleation 2	[82]
Germany, India, Mexico, France	2	*Klebsiella pneumoniae* 1 Other 1	Other 1	Lung diseases 1 Immunosuppression 1 Other 2	2 (RE 1 LE 1)	Monocular 2	Vitrectomy 2	[83]
India	1	*Escherichia coli* 1	Other 1	Diabetes mellitus 1	2 (RE 1 LE 1)	Binocular 1	No surgery	[84]
China	1	*Klebsiella pneumoniae* 1	BSI/Sepsis 1	Diabetes mellitus 1 Other 1	1 (LE 1)	Monocular 1	Vitrectomy 1	[85]
Portugal	1	*Klebsiella pneumoniae* 1	Liver abscess 1	NA	1 (RE 1)	Monocular 1	Vitrectomy 1	[86]
USA	1	*Klebsiella pneumoniae* 1	NA	Diabetes mellitus 1 Drug abuse 1	1 (RE 1)	Monocular 1	No surgery	[87]
USA	1	*Klebsiella pneumoniae* 1	Liver abscess 1	Chronic hepatitis 1	1 (LE 1)	Monocular 1	No surgery	[88]
India	1	Other 1	Other 1	NA	1 (RE 1)	Monocular 1	Vitrectomy 1	[89]
India	1	*Klebsiella pneumoniae* 1	Other 1	NA	1 (LE 1)	Monocular 1	Enucleation 1	[90]
Japan	2	*Klebsiella pneumoniae* 1	Liver abscess 2	Diabetes mellitus 1 Other 1	2 (RE 2)	Monocular 2	Vitrectomy 3 Enucleation 1	[91]
United Kingdom	1	*Escherichia coli* 1	Other 1	Gastrointestinal disorders 1 Hypertension 1	1 (LE 1)	Monocular 1	NA	[92]
China	1	*Klebsiella pneumoniae* 1	NA	Diabetes mellitus 1 Lung disease 1	1 (RE 1)	Monocular 1	Vitrectomy 1 Enucleation 1	[93]
India	1	*Klebsiella pneumoniae* 1	NA	Chronic hepatitis 1	1 (LE 1)	Monocular 1	NA	[94]
USA	1	*Klebsiella pneumoniae* 1	Liver abscess 1	Diabetes mellitus 1	1 (LE 1)	Monocular 1	No surgery	[95]
China	1	*Klebsiella pneumoniae* 1	Liver abscess 1	Diabetes mellitus 1	1 (RE 1)	Monocular 1	Vitrectomy 1	[96]
Canada	1	*Klebsiella pneumoniae* 1	Other 1	Diabetes mellitus 1 Renal disease 1	1 (LE 1)	Monocular 1	Vitrectomy 1	[97]
China	1	*Klebsiella pneumoniae* 1	Other 1	Diabetes mellitus 1 Hypertension 1	1 (LE 1)	Monocular 1	Vitrectomy 1	[98]
India	1	*Klebsiella pneumoniae* 1	Other 1	None 1	1 (LE 1)	Monocular 1	No surgery	[99]
South Korea	1	*Klebsiella pneumoniae* 1	Liver abscess 1	None 1	1 (RE 1)	Monocular 1	Vitrectomy 1	[100]
South Korea	1	*Klebsiella pneumoniae* 1	Liver abscess 1	None 1	1 (RE 1)	Monocular 1	Vitrectomy 1	[100]
India	1	Other 1	BSI/Sepsis 1	None 1	1 (LE 1)	Monocular 1	Vitrectomy 1 Evisceration 1	[101]
China	1	*Klebsiella pneumoniae* 1	NA	Alcohol abuse 1 Gastrointestinal disorders 1	1 (RE 1)	Monocular 1	Vitrectomy 1	[102]
Australia	4	*Klebsiella pneumoniae* 4	Liver abscess 4 Pneumonia 1 BSI/Sepsis 3 Other 1	Dyslipidaemia 3 Diabetes mellitus 2 Hypertension 1 None 1	6 (RE 3 LE 3)	Monocular 2 Binocular 2	Vitrectomy 4 Enucleation 1	[103]
France	1	*Klebsiella pneumoniae* 1	Liver abscess 1	None 1	1 (LE 1)	Monocular 1	No surgery	[104]
USA	1	*Klebsiella pneumoniae* 1	BSI/Sepsis 1	Diabetes mellitus 1	1 (RE 1)	Monocular 1	Enucleation 1	[105]
Australia	1	*Escherichia coli* 1	NA	Diabetes mellitus 1	1 (LE 1)	Monocular 1	Vitrectomy 1	[106]
Saudi Arabia	1	Other 1	Other 1	NA	1 (LE 1)	Monocular 1	Vitrectomy 3	[107]
United Kingdom	1	Other 1	NA	Renal diseases 1	1 (LE 1)	Monocular 1	No surgery	[108]
United Kingdom	2	*Klebsiella pneumoniae* 2 *Pseudomonas aeruginosa* 1	Pneumonia 1 Other 1	Diabetes mellitus 1 Renal diseases 1	3	Monocular 1 Binocular 1	NA	[109]
China	1	*Klebsiella pneumoniae* 1	Liver abscess 1	Diabetes mellitus 1	1 (LE 1)	Monocular 1	Vitrectomy 1	[110]
Australia	1	*Klebsiella pneumoniae* 1	Liver abscess 1	Dyslipidaemia 1	2 (RE 1 LE 1)	Binocular 1	Vitrectomy 1	[111]
China	1	Other 1	NA	NA	1 (LE 1)	Monocular 1	Vitrectomy 1	[112]
China	1	*Klebsiella pneumoniae* 1	Pneumonia 1 BSI/Sepsis 1	Other 1 Chronic hepatitis 1	2 (RE 1 LE 1)	Binocular 1	No surgery	[113]
India	1	*Klebsiella pneumoniae* 1	Other 1	Pregnancy complications 1	1 (LE 1)	Monocular 1	No surgery	[114]
India	6	*Klebsiella pneumoniae* 2 *Pseudomonas aeruginosa* 2	NA	NA	NA	NA	NA	[115]
Japan	1	*Klebsiella pneumoniae* 1	BSI/Sepsis 1	None 1	2 (RE 1 LE 1)	Binocular 1	No surgery	[116]
Japan	1	*Klebsiella pneumoniae* 1	Other 1	Other 2	1 (LE 1)	Monocular 1	Vitrectomy 1 Enucleation 1	[117]
United Kingdom	1	Other 1	Other 1	None 1	1 (RE 1)	Monocular 1	No surgery	[118]
Saudi Arabia	1	*Klebsiella pneumoniae* 1	Liver abscess 1	Diabetes mellitus 1	1 (RE 1)	Monocular 1	Vitrectomy 1 Evisceration 1	[119]
Thailand	1	*Klebsiella pneumoniae* 1	BSI/Sepsis 1	Diabetes mellitus 1	1 (LE 1)	Monocular 1	Vitrectomy 1	[120]
Saudi Arabia	2	*Klebsiella pneumoniae* 2	Liver abscess 2	Diabetes mellitus 2 Cardiovascular disease 1 Hypertension 1 Lung diseases 1	3 (RE 1 LE 2)	Monocular 1 Binocular 1	Vitrectomy 1	[121]
Croatia	1	*Pseudomonas aeruginosa* 1	BSI/Sepsis 1	Hypertension 1 Cardiovascular disease 1	2 (RE 1 LE 1)	Binocular 1	No surgery	[122]
Canada	1	*Escherichia coli* 1	NA	Hypertension 1	1 (RE 1)	Monocular 1	Vitrectomy 1	[123]
Taiwan	2	*Klebsiella pneumoniae* 1	Other 2	Diabetes mellitus 2	3 (RE 2 LE 1)	Monocular 1 Binocular 1	No surgery	[124]
Nepal	2	*Escherichia coli* 2	NA	NA	2 (RE 2)	Monocular 2	NA	[125]
Taiwan	1	*Citrobacter* spp. 1	Other 1	NA	1 (RE 1)	Monocular 1	NA	[126]
USA	1	*Klebsiella pneumoniae* 1	Liver abscess 1	None 1	1 (RE 1)	Monocular 1	Enucleation 1	[127]
Spain	1	Other 1	Other 1	Diabetes mellitus 1 Other 1	1 (LE 1)	Monocular 1	Evisceration 1	[128]
Belgium	1	*Klebsiella pneumoniae* 1	Liver abscess 1	Diabetes mellitus 1	1 (RE 1)	Monocular 1	No surgery	[129]
Thailand	1	*Pseudomonas aeruginosa* 1	Other 1	None 1	1 (LE 1)	Monocular 1	No surgery	[130]
Hong Kong	1	*Pseudomonas aeruginosa* 1	NA	Lung diseases 1	1 (RE 1)	Monocular 1	Enucleation 1	[131]
United Kingdom	1	*Pseudomonas aeruginosa* 1	Other 1	NA	2 (RE 1 LE 1)	Binocular 1	No surgery	[132]
Taiwan	1	Other 1	NA	Diabetes mellitus 1	1 (RE 1)	Monocular 1	No surgery	[133]
Thailand	1	Other 1	BSI/Sepsis 1	Other 1	1 (LE 1)	Monocular 1	No surgery	[134]
Spain	1	*Pseudomonas aeruginosa* 1	Pneumonia 1	None 1	1 (RE 1)	Monocular 1	Enucleation 1	[135]
Australia	1	*Klebsiella pneumoniae* 1	Liver abscess 1	None 1	1 (RE 1)	Monocular 1	Enucleation 1 Evisceration 1	[136]
Malaysia	1	*Klebsiella pneumoniae* 1	Other 1	Diabetes mellitus 1 Malignancy 1	1 (LE 1)	Monocular 1	Evisceration 1	[137]
USA	1	*Pseudomonas aeruginosa* 1	Other 1	Lung diseases 1 Immunosuppression 1	2 (RE 1 LE 1)	Binocular 1	Vitrectomy 1	[138]
Japan	1	*Klebsiella pneumoniae* 1	Other 1	Diabetes mellitus 1	1 (RE 1)	Monocular 1	No surgery	[139]
USA	1	*Pseudomonas aeruginosa* 1	Pneumonia 1	Lung diseases 1	1 (LE 1)	Monocular 1	Vitrectomy 1	[140]
Ireland	1	*Pseudomonas aeruginosa* 1	BSI/Sepsis 1	Pregnancy complications 1	1 (RE 1)	Monocular 1	Evisceration 1	[141]
China	1	*Pseudomonas aeruginosa* 1	NA	Lung diseases 1	1 (RE 1)	Monocular 1	Vitrectomy 1	[142]
Slovakia	1	*Pseudomonas aeruginosa* 1	BSI/Sepsis 1	Pregnancy complications 1	2 (RE 1 LE 1)	Binocular 1	No surgery	[143]
Japan	1	*Klebsiella pneumoniae* 1	Liver abscess 1	None 1	2 (RE 1 LE 1)	Binocular 2	Enucleation 2	[144]
USA	1	Other 1	Other 1	Other 1	2 (RE 1 LE 1)	Binocular 1	Vitrectomy 2	[145]
Taiwan	1	*Klebsiella pneumoniae* 1	NA	Renal disease 1	1 (RE 1)	Monocular 1	No surgery	[146]

N° Eyes (n° RE, n° LE) = number of eyes involved (number of right eyes involved, number of left eyes involved); BSI/Sepsis = Bloodstream infection/Sepsis.

**Table 2 microorganisms-11-00080-t002:** Comorbidities in the selected population.

Comorbidities	
Number of studies with available data	84
Number of patients included	420
Diabetes mellitus (n, %)	231 (55.0%)
Hypertension (n, %)	72 (17.1%)
Renal disease (n, %)	21 (5.0%)
Malignancy (n, %)	19 (4.5%)
Chronic hepatitis (n, %)	18 (4.3%)
Cirrhosis (n, %)	17 (4.0%)
Gastrointestinal disorders (n, %)	14 (3.3%)
Lung diseases (n, %)	11 (2.6%)
Other Cardiovascular diseases (n, %)	7 (1.7%)
Immunosuppression (n, %)	5 (1.2%)
Alcohol abuse (n, %)	5 (1.2%)
Pregnancy complications (n, %)	4 (0.9%)
People who inject drugs (n, %)	3 (0.7%)
Other (n, %)	29 (6.9%)

**Table 3 microorganisms-11-00080-t003:** Source of infection.

Source of Infection	
Number of studies with available data	81
Number of patients included	488
Liver abscess (n, %)	266 (54.5%)
Bloodstream infections/sepsis (n, %)	116 (23.8%)
Pneumonia (n, %)	37 (7.6%)
Abdominal infections (n, %)	37 (7.6%)
Urinary tract infections (n, %)	26 (5.3%)
Acute bacterial skin and skin structures infections (n, %)	8 (1.6%)
Post-surgery or medical procedures (n, %)	8 (1.6%)
Other source of infection (n, %)	19 (3.9%)

**Table 4 microorganisms-11-00080-t004:** Initial ocular lesions.

Initial Ocular Lesions	
Number of studies with available data	70
Number of eyes included	270
Vitreous opacity (n, %)	134 (49.6%)
Hypopyon (n, %)	95 (35.2%)
Redness (n, %)	91 (33.7%)
Corneal involvement (n, %)	66 (24.4%)
Anterior chamber reaction (n, %)	46 (16.7%)
Swelling lid (n, %)	37 (13.7%)
Retinal lesions (n, %)	27 (10.0%)

## Data Availability

Not applicable.

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
