# Peer review of "Gram-Negative Endogenous Endophthalmitis: A Systematic Review"

_microorganisms, 2022, doi:10.3390/microorganisms11010080_

Round 1

Reviewer 1 Report

I have revised the manuscript, and here are my comments:
The scientific names must be italic such as in lines 30 and 44; check throughout the manuscript
Line 43 SPP not italic
Sec 2.3. rephrase for clarity, and sentences’ structure as (AS, DL, MM, AM, SS, GT) should be added before the last word
Enhance Figure 1 and Figure 2 resolution
The table caption should be before the table
I suggest redesigning Table 1 for clarity no need for the first three columns; delete, and move the Reference column to the last column in the table; then, in the footnote, add all abbreviations in the table.
Enhance the discussion part.
Check the linguistic mistakes in the manuscript
Delete “authors” from “authors' conclusion.”
Check the outputs of all references also, the style
Again, scientific names are not italic in the references, check

Reviewer 2 Report

In this review, authors describe endogenous bacterial endophthalmitis (EBE) caused by gram-negative bacteria majority by Klebsiella pneumoniae, Pseudomonas aeruginosa and Escherichia coli, explored their antimicrobial resistance rate and outcomes of EBE.

The manuscript is well-written and applied methods are appropriate.

This reviewer noticed following points to be addressed.

1.       In abstract, EBE should be expanded as the first-time appearance in the manuscript.

2.       Line 152-153, “A total of 591 patients were included, prevalently Asia (98, 81.7%), male (302, 62.9%)”, is it correct? check and confirm (%)

3.       Line 157-162, check the (% ) of prevalence of aetiologic agents and total patients 591 or 592?

4.       Please confirm the number of total infected eyes (n=?) which should be described in the table as well. Check the number of overall infected eyes and clarify in the text.

5.       Line 280-282, rephrase the sentence of “A recent study reported that……………..Pseudomonas spp.(92%).” because Bacillus spp. are Gram-positive bacilli not Gram-negative pathogens.

6.       Organism names should be italicized throughout the manuscript.

Round 2

Reviewer 1 Report

The authors have carefully processed all comments. The quality of the manuscript has increased significantly. I have no further comments.